# Brighton v Will: The Legal Chasm between Animal Welfare and Animal Suffering

**DOI:** 10.3390/ani10091497

**Published:** 2020-08-24

**Authors:** Sophie Riley

**Affiliations:** Faculty of Law, University of Technology Sydney, PO Box 123, Broadway, Sydney, NSW 2007, Australia; sophie.riley@uts.edu.au

**Keywords:** animal welfare, animal cruelty, utilitarianism, statutory interpretation, contextual interpretation

## Abstract

**Simple Summary:**

Animal welfare regulation forms the backbone of regimes designed to prevent cruelty to animals. The welfare concept is based on animal sentience—the notion that animals feel pain and also suffer. Accordingly, legislation aims to cushion animals from cruelty, but also balances this against the needs of humans. While dogs are popularly described as a person’s best friend, in law they can also be classified as pests. As the case of *Brighton v Will* demonstrates, a literal (textual) interpretation of legislation can justify a high degree of animal cruelty for animals classified as a pest. In reality, judges who interpret legislation can widen or narrow the reach of anti-cruelty provisions according to their interpretation. In particular, where regimes comprise a mixture of legislation and extrinsic documents such as codes, plans and strategies, which add light and shade to legislation, judges should interpret legislation according to context, established by these instruments (a contextual interpretation). This allows the law to adjust in a timely way to societal and policy developments regarding prevention of cruelty to animals.

**Abstract:**

Through the mechanism of statutory interpretation, courts can narrow or widen the legal concept of animal cruelty. This was starkly brought to light in the case of *Brighton v Will*, where the Supreme Court of New South Wales held that stabbing a dog six times with a pitchfork and then killing him with a mallet, did not amount to serious animal cruelty. This finding was the result of the Court’s applying a textual interpretation to the NSW Crimes Act, concluding that the appellant was simply exterminating a pest. Yet, animal law in NSW comprises more than legislation, extending to a raft of plans and strategies which provide background and context for regulation. This article argues that a contextual interpretation would have been more appropriate, leading to enquiries whether the dog was rightfully classified as a pest, as well as whether the law should have considered the manner in which the dog was killed. An equally relevant issue stems from the relationship between animal suffering and animal welfare, a connection which hinges on the ambit of anti-cruelty legislation. The latter permits a range of exceptions and defences that permit justification of cruelty, magnifying the chasm between animal suffering and animal welfare. This chasm is also not diminished by legal interpretations of cruelty that focus on whether killing is justified, while ignoring the method of killing.

## 1. Introduction

On 3 April 2020, the Supreme Court of New South Wales (the Supreme Court) handed down its decision in the case of *Brighton v Will* (*Brighton v Will* (2020) NSWSC 435 [1]). The decision was an appeal from a conviction in the Local Court, where pursuant to Section 530 of the *Crimes Act 1900* (NSW) (Crimes Act [2]), the appellant, Brighton, had been found guilty of serious animal cruelty.

Brighton owned a camel named Alice who had been attacked by a dog. In defending Alice, Brighton killed the dog by stabbing him six times with a pitchfork and then, while the dog was still alive, suspended him from a tree, finally killing the dog by hitting him six or eight times with a mallet (*Brighton v Will*, parags 15–16 [1]). The Supreme Court acknowledged that the method of killing was particularly cruel, but nevertheless reversed the decision of the Local Court, holding that in accordance with Section 530(2) of the Crimes Act, the appellant had exterminated a pest (*Brighton v Will*, parag 117 [1]). The Supreme Court’s decision turned on the seemingly straightforward proposition, that as a matter of statutory interpretation, the words “pest animal” and “extermination” should be given their ordinary meaning. Yet, the result appears to side step one of purposes of anti-cruelty regulation, which is to punish ill treatment of animals.

The aim of this article is to analyse the Supreme Court’s decision, offering two arguments: first, that a contextual, rather than textual, approach to statutory interpretation would have been more appropriate; and second, that justifying animal cruelty in cases of abject cruelty magnifies the legal chasm between animal welfare and animal suffering. The two arguments are linked through the mechanism of anti-cruelty regulation, which does not proscribe cruelty as such, but only cruelty which is unreasonable, unnecessary or unjustified (*Prevention of Cruelty to Animals Act 1979* (NSW), s 4(2) [3]).

With respect to the first argument, pest animal management in New South Wales (NSW) comprises a range of instruments, such as codes, strategies and management plans, which are extrinsic to legislation [4,5,6,7,8,9,10]. The complexity and interconnectedness of this corpus of material make it challenging to interpret legislative terms solely on the basis of dictionary meanings. In particular, a dictionary definition may be unresponsive to regulatory bearings, such as the requirement that killing should be carried out humanely]. With respect to the second argument, the chasm between animal welfare and animal suffering derives from the utilitarian underpinnings of animal welfare, but it can also be aggravated by the way courts construe words and terms, focusing on whether killing is justified but condoning cruelty by overlooking the method of killing.

The discussion commences with a description and analysis of the decision in *Brighton v Will* [1], before moving to rules of statutory interpretation and an evaluation of how contextual interpretation could have led to a different result. The analysis then examines the chasm between animal welfare and animal suffering, concluding that the complexity of regulatory regimes calls for greater flexibility in construing statutes that allow courts to have recourse to a greater range of materials for enhanced interpretive insights.

Before commencing the discussion, the reader may have noticed that references to the dog avoid using the pronoun “it”, instead referring to the dog as “he” or “him”. This use of the third person pronoun is deliberate. It has been done in recognition of the inherent value of animals, and is a configuration used throughout this paper, including the feminine versions “she” and “her”, where relevant and when referring to other animals. Finally, animal welfare is not currently defined in legislation in New South Wales and, for the purpose of this article, the definition adopted is the one developed by the World Organisation for Animal Health (OIE) [11] as “the physical and mental state of an animal in relation to the conditions in which it lives and dies. An animal experiences good welfare if the animal is healthy, comfortable, well nourished, safe, is not suffering from unpleasant states such as pain, fear and distress and is able to express behaviours that are important for its physical and mental state” [12].

## 2. Brighton v Will

### 2.1. The Facts and the Appeal

The *Prevention of Cruelty to Animals Act 1979* (NSW) (POCTA) [3] and the Crimes Act [2] both contain offences for animal cruelty. Section 5 of POCTA [3] prohibits cruelty, Section 6 prohibits aggravated cruelty, and Section 530 of the Crimes Act [2] prohibits serious animal cruelty. The latter was added to the Crimes Act by the *Crimes Amendment (Animal Cruelty) Act 2005* NSW for the purpose of punishing significant animal cruelty and proffers stricter penalties than those specified in POCTA [13]. Depending on the severity of the cruelty, a prosecutor may therefore determine that there are advantages in bringing an action under the Crimes Act. Table 1 sets out penalties for breaches of each section.

In *Brighton v Will*, the Royal Society for the Prevention of Cruelty to Animals (RSPCA) commenced action against Brighton under the Crimes Act, due to the serious nature of the acts of cruelty (*Brighton v Will*, parag 1 [1]). Section 530 states that:
(1)A person who, with the intention of inflicting severe pain--
(a)tortures, beats or commits any other serious act of cruelty on an animal, and(b)kills or seriously injures or causes prolonged suffering to the animal, is guilty of an offence.

Maximum penalty: imprisonment for 5 years. 

(1A)A person who, being reckless as to whether severe pain is inflicted--
(a)tortures, beats or commits any other serious act of cruelty on an animal, and(b)kills or seriously injures or causes prolonged suffering to the animal, is guilty of an offence.

Maximum penalty: imprisonment for 3 years. 

(2)A person is not criminally responsible for an offence against this section if--
(a)the conduct occurred in accordance with an authority conferred by or under the *Animal Research Act 1985* or any other Act or law, or(b)the conduct occurred in the course of or for the purposes of routine agricultural or animal, husbandry activities, recognised religious practices, the extermination of pest animals, or veterinary practice.


As indicated in the introduction to this article, Brighton owned a camel named Alice and she was part of a mobile petting zoo, which provided Brighton with income. On 16 January 2016, two dogs attacked Alice, injuring her (*Brighton v Will*, parag 11 [1]). Brighton captured one of the dogs, who by this time had become docile and tied the dog to a tree. By “docile” the court meant that “the dog no longer posed a threat to the camel (nor anyone or anything else) because it was tied up and subdued” (*Brighton v Will*, parag 44 [1]).

Again as mentioned in the introduction, Brighton then “stabbed the dog at least six times with a pitchfork” and left him tied to the tree while he sought veterinary assistance for Alice (*Brighton v Will*, parags 12–14). When Brighton returned, he realised the dog was still alive and “suspended the dog from a tree and beat it across the head between 6 and 8 times, with a mallet,” eventually killing the dog, and burying it (*Brighton v Will*, parags 12–14 [1]). 

In the Local Court, Brighton relied on Section 530(2)(b) of the Crimes Act, arguing that he was exterminating a pest. The Magistrate disagreed, holding that the dog was neither a “pest”, nor the killing an “extermination”, convicting Brighton of animal cruelty and sentencing him to three years and four months imprisonment (*Brighton v Will*, parags 9, 90, 177–183, 186 [1]). Brighton appealed both the conviction and the sentence, by way of summons to the Supreme Court.

The facts of the case were not in dispute, and his Honour Rothman J acknowledged that the dog was killed in a “particularly abhorrent” and cruel manner, especially as the dog was already subdued at the time of the stabbing and subsequent killing (*Brighton v Will*, parag 130 [1]). The legal question, however, was not whether the conduct was reprehensible, but whether it was criminal. (*Brighton v Will*, parag 130.) Rothman J further noted that there were a number of grounds of appeal, all of which centred on the core argument that the Magistrate had misinterpreted Section 530(2) of the Crimes Act, particularly Section 530(2)(b) (*Brighton v Will*, parag 89 [1]). There was some discussion as to whether the latter section provided an actual defence, but it was not a matter that needed to be decided because it was not raised in argument. (*Brighton v Will*, parag 43.) Although his Honour alluded to the section as a “so called defence”, for ease of reference, the section will be referred to as a defence in this article (*Brighton v Will*, parag 68 [1]).

### 2.2. Interpretation of “Pest” and “Exterminate”

Notwithstanding the fact that they are critical to an adjudication of serious animal cruelty, neither the word “pest”, nor the word “exterminate” is defined in the legislation, raising complex issues of statutory interpretation (*Brighton v Will*, parag 71 [1]).

In the Local Court, the Magistrate had construed these words by interpreting them in the broader context of pest animal regulation in New South Wales (NSW) (*Brighton v Will*, parag 48 [1]). Accordingly, the Magistrate directed attention to instruments extending beyond the Crimes Act, such as the Local Land Services (Wild Dogs) Pest Control Order 2015 (NSW) (Wild Dog Pest Control Order 2015) [14], the *Companion Animals Act 1998* (NSW) [15] (Companion Animals Act), the *Exhibited Animals Protection Act 1986* (NSW) and the *Animal Research Act 1985* (NSW) (*Brighton v Will*, parag 48 [1]). As already noted, in using these instruments, the Magistrate found that the dog was not a “pest”, nor the killing an “extermination” (*Brighton v Will*, parags 9, 90, 177–183, 186 [1]). The appellant, however, submitted that this amounted to an error of law, because the words were given a technical meaning, rather than their ordinary meaning (*Brighton v Will*, parag 71 [1]).

In the Supreme Court, Rothman J held that whether a word should be given a technical or ordinary meaning is a question of law. Nevertheless, if a court determines that words should be given their ordinary meaning, that meaning itself is a question of fact (*Brighton v Will*, parag 25 [1]). Accordingly, if the Magistrate should have given the words an ordinary meaning, rather than a technical meaning, this would amount to an error of law, which was the very argument of the appellant (*Brighton v Will*, parag 47 [1]).

In this regard, Brighton contended that whether the dog was a pest depended on the dog’s conduct, especially the threat posed to Alice at a specific point in time (*Brighton v Will*, parag 34 [1]). Moreover, the fact that the dog had stopped attacking Alice and was comparatively docile at the time he was killed did not change the fact that dog could still objectively be classified as a pest. (*Brighton v Will*, parags 34, 51.) The respondent, on the other hand, argued that the dog was not a pest because he was microchipped, was wearing a collar and was “subdued and submissive upon restraint” (*Brighton v Will*, parag 56 [1]). In addition, the fact that Brighton had tied up the dog (with pitchfork still embedded) meant that at this point the dog was no longer a threat (*Brighton v Will*, parag 58 [1]). The respondent further argued that in interpreting the word “exterminate”, the method of killing was relevant, particularly as the threat to Alice no longer existed at the time of killing and the way the dog was killed was exceedingly cruel (*Brighton v Will*, parags 86, 116 [1]). In answer, the appellant contended that the overall intention of Section 530, particularly in view of the defence created by Section 530(2), was to forgive methods of killing which under other conditions would breach Section 530(1); hence, the method of killing was not in issue (*Brighton v Will*, parag 73 [1]). Furthermore, to interpret the section in any other way, especially adjudicating on the method of killing, would render the defence provided by Section 530(2) futile (*Brighton v Will*, parag 73 [1]).

After considering these arguments, the Court found that, as a matter of statutory interpretation, the terms “pest animal” and “extermination” were to be given their ordinary meaning, so that a pest animal was an animal who could be described as “noxious, destructive or troublesome…; a nuisance animal”, while “extermination” meant “to destroy utterly (especially something living)” or “get rid of, eliminate (a pest, disease)” (*Brighton v Will*, parags 72, 135–136 [1]). His Honour pointedly noted that the dog adequately fitted the description of a pest and the appellant did no more or less than destroy the dog (*Brighton v Will*, parag 72 [1]). In taking this approach, Rothman J removed the context that the Magistrate had considered important, finding that the appellant was justified, not only in killing the dog, but killing him in a way that would otherwise have amounted to serious animal cruelty (*Brighton v Will*, parag 117 [1]).

Yet, given that Section 530 was introduced specifically to deal with serious animal cruelty, it seems incongruous that the method of killing could be disregarded. The next part argues that a contextual approach to statutory interpretation would have better identified parliament’s intention in amending the Crimes Act.

## 3. Text or Context?

### 3.1. Rules of Statutory Interpretation

Statutory interpretation has been described as the task of identifying “the crystallized meaning [of terms] by application in decisions in actual cases” (page 229 [16]). Yet, such meaning may not always be “plain and explicit”, so statutes must be interpreted (page 229 [15]). One approach is to apply a literal reading, identifying the words’ plain or ordinary meaning. In Australia, this was popular until the 1970s–1980s, when judges and commentators urged caution, pointing to the fact that statutory interpretation is an art rather than a science (pages 113, 115 [17]). Accordingly, the aim should be to interpret legislation in its entire context, an encouragement which also raises important issues concerning how far context extends. (page 322 [18], page 115 [17], page 41 [19], page 1084 [20])

Barnes argues that contextual interpretation involves construing words and phrases in relation to other statutes, provisions and extrinsic material, so that words and phrases are viewed as an “integral part of an organic whole”, rather than in isolation (page 1105 [20]). This is consistent with the High Court’s standpoint, that while interpretation must start with the text.

The modern approach to statutory interpretation (a) insists that the context be considered in the first instance, not merely at some later stage when ambiguity might be thought to arise, and (b) uses “context” in its widest sense to include such things as the existing state of the law and the mischief which, by legitimate means such as those just mentioned, one may discern the statute was intended to remedy (page 408, [21]).

This statement clearly envisages that text, purpose and context are linked, suggesting a broad ambit for contextual interpretation (pages 402, 416 [22]). Pearce and Gedde’s *Statutory Interpretation in Australia* adopts an analogous approach, maintaining that context tempers the plain and ordinary meaning of words, going as far as suggesting that the “formal effect” of s 15AA of the Acts Interpretation Act 1901 (Cth) and their equivalent at the state level, “is to override both the common law literal approach… and the purposive approach to interpretation”, so that, from the outset, words need to be interpreted in context (page 60 [19]).

A number of rules of statutory interpretation already go some way towards considering context, such as when statues are pari materia, in that they share a common purpose and are therefore construed together (page 60 [19]). This rule, however, does not apply simply because the same words or terms are used in a variety of statutes. If the statutes do not share a common purpose, a court is not bound by an interpretation given to identical words in different legislation (page 9 [19]). Another way of integrating context comes from linking statutes that may not be pari materia, but form a scheme for regulation. If there is a “rational integration” of legislation, courts construe the statutes to produce a sensible result (page 131 [19]). Yet a further way of incorporating context derives from the maxim that the statute is always speaking. This allows the law to remain up to date, rather than being constrained by interpretations dating from the time the statute commenced (page 157 [19]).

The Privy Council and Australian courts have decided cases on the basis of each of these approaches, although this has not led to the development of overarching principles (page 726 [23]; page 287 [24,25,26]; pages 128, 131, 156–157 [19]). Arguably, even if such principles had been developed, they would not have been entirely helpful in Brighton v Will [1], because the regime under consideration comprises a significant number of extrinsic instruments, such as strategies, plans, and codes, which contain detail and operational processes [5,6,7,8,9,10]. While these instruments provide context, questions concerning their relevance and admissibility remain. The issue is all the more difficult, because of the volume and complexity of these instruments and the fact that they deal with different categories of dogs.

### 3.2. Context: Dogs as Companion Animals and Pests

Dogs are popularly known as a person’s best friend, but this epithet doubtlessly applies to well-behaved companion animals kept as domestic pets. Since 1998, the Companion Animals Act has established a regime for the care and management of domestic pets, such as cats and dogs, including registration of ownership (Sections ss 3A, 9, 10) [15]. Moreover, the anti-cruelty provisions of POCTA apply to companion dogs as much as other animals (Sections 5, 6, 8, 9, 11, 14, 15, 18 [3]). However, dogs not classified as companion animals occupy a precarious position in society.

As early as 1830, laws in NSW authorised the killing of stray dogs, attracting a bounty of two shillings and six pence (Sections 1, 2 [27]). Dingoes were singled out as agricultural pests from the mid nineteenth century (Sections 1, 3 [28]); and the Wild Dog Destruction Act 1921 (NSW) regulated the eradication of dingoes and wild dogs in the Western Division of NSW, until the enactment of the Local Land Services Act 2013 (NSW) (LLS Act) (Sections 1, 3, 4 [29,30]). Amongst other things, the LLS Act created eleven Local Land Services Boards charged with providing agricultural advisory services and managing natural resources and biosecurity (Sections 4, 6, 14, Schedule 1 [30]). In accordance with the LLS Act, the NSW government gazetted the Wild Dog Pest Control Order 2015, which declared wild dogs to be a pest and obliged occupiers of property to control wild dogs “by any lawful method”(clause 7(a) [15]. It is expected that eventually the pest control order will be phased out and replaced by Local Land Services Plans, which deal with pest animals on a local basis, but which are consistent with the Biosecurity Act 2015 (NSW) (Biosecurity Act) which aims to prevent, eliminate and minimise biosecurity risks in NSW, including eradicating new pests and managing pests that cannot be eradicated (Section 3 [31]).

The evolution of the regime for destruction of wild dogs in NSW pre-dates laws dealing with animal cruelty. While stray dogs were targeted for destruction from 1830, the NSW Parliament passed its first anti-cruelty laws in 1850, copying the *Cruelty to Animals Act 1849* (UK) verbatim [32]. Early eradication techniques were not subject to many limitations and authorised the use of strychnine (Section 1, Schedule 1 [33], which was declared cruel in at least one case in the mid-twentieth century (page 161 [34]) and is now not considered a humane way of killing (Appendix D [5]. Indeed, one of the problems facing those who interpret legislation relating to pest species is that the regulatory milieu has changed substantially, particularly in the last two decades. During that time, the NSW Parliament has amended the Crimes Act, as well as having passed wide-ranging statutes, such as the LLS Act and the Biosecurity Act. The last two pieces of legislation are supported by a raft of policy instruments, orders, strategies and plans, including many specifically relating to wild dog management [5,6,7,8].

The regime deals with different types of dogs in different ways, although the concept of a wild dog as a pest weaves its way through many of the instruments. From the field of environmental protection, “Predation and hybridisation by Feral Dogs, *Canis lupus familiaris*” is listed as a key threatening process pursuant to the *Biodiversity Conservation Act 2016* (NSW) (Section 4.31, Schedule 4 [35]). The *New South Wales Invasive Species Plan 2018–2021*, regards “invasive species” as equivalent to “pest” species, singling out wild dogs as one of the “most significant (and) widespread pest animals in NSW” (page 5 [8]). Furthermore, the definition of invasive species in the plan “species whose establishment and spread threatens ecosystems, habitats or species with economic or environmental harm”, is consistent with the identification of wild dogs as pests pursuant to the Wild Dog Pest Control Order 2015 (page 5, Appendix 2, Glossary [14]). Finally, dogs are managed as pests through the *Model Code of Practice for the Humane Control of Wild Dogs* (2013) which assumes that wild dogs are the equivalent of pest animals:
“native or introduced, wild or feral, non-human species of animal that…(are) currently troublesome locally, or over a wide area, to one or more persons, either by being a health hazard, a general nuisance, or by destroying food, fibre, or natural resources” (page 1 [5]).

In addition to these references, the word pest is also found elsewhere in legislation. Section 15(1) of the Biosecurity Act, for example, contains a definition of a pest [31]. However, courts are unable to use this definition to interpret Section 530 of the Crimes Act because the two statutes have differing objectives. The Biosecurity Act establishes frameworks for stakeholders to deal with biosecurity risks (Section 3 [31]) while the object of Section 530 of the Crimes Act is to create an offence for serious animal cruelty [13,36]. In the absence of common purposes, transposing definitions in these statutes would be counter-productive.

However, as discussed, the management of wild dogs comprises more than statutory directives. Legislative and non-legislative instruments operate as a whole, providing context for the regime and influencing the meaning of words and phrases beyond dictionary definitions. In *Brighton v Will*, the dictionary definition of a pest, “noxious, destructive or troublesome animal; a nuisance animal” (*Brighton v Will*, parags 135–136 [1]) could just as easily describe a badly-behaved pet dog. For this reason, the Wild Dog Pest Control Order 2015 excludes dogs kept pursuant to the Companion Animals Act, or dogs kept for exhibition or research purposes (clause 3, definitions [14]). Consequently, if the dog in *Brighton v Will* [1], had been acknowledged to be a domestic dog, arguably the dictionary definition of a pest would not have applied to him. Instead, he would have been dealt with pursuant to the Companion Animals Act. The dog could have been declared a dangerous dog, and if he had to be euthanized, this would have occurred in a way that caused him “to die quickly and without unnecessary suffering” (Section s 48(7) [15]). Even if the dog had not been declared dangerous and the appellant had killed the dog in accordance with Section 22 of the Companion Animals Act, the killing should not have involved unnecessary suffering [15].

An analogous challenge derives from the fact that words and phrases may have attained a meaning, known as a “specific” meaning, but which falls short of a technical meaning (page 167, [19]). The Court in *Brighton v Will* emphasised that giving words a technical meaning where it is not warranted amounts to an error of law (*Brighton v Will*, parag 71 [1]). However, in *Canonba Pastures Protection Board v Leigh* (the Canoba case), the court held that a word can acquire a special meaning, which can modify a dictionary definition [37]. In that case, a proclamation had declared feral pigs to be a noxious animal throughout NSW, triggering obligations on the part of occupiers to control the pigs and/or allow officers of the Canonba Pastures Protection Board to destroy them (parag 2 [37]. Leigh had captured a number of wild pigs, some of whom had produced offspring in captivity, leading Leigh to argue that the pigs and their offspring were not feral. Had this argument succeeded, he could effectively have disputed that the Canonba Pastures Protection Board had authority to destroy the pigs (parag 3 [37]). The issues before the Court, therefore, turned on the meaning of the phrase “feral pig”.

In the Local Court, the Magistrate determined that the term “feral” was not a technical term, and proceeded to give the word its ordinary meaning as “wild or untamed” (parag 4 [37]). This resulted in a finding that at the time the pigs were shot, they were domestic pigs. In coming to this decision, the Magistrate refused to admit expert testimony regarding physical characteristics of wild pigs, relating to the shape and size of their snout and bristles (parag 8 [37]). On appeal to the Supreme Court, Priestly JA held that this part of the decision was in error, citing with approval the case of *D & R Henderson v Collector of Customs*, 48 ALJR 132, where the High Court determined that a word which did not have a technical meaning, could nevertheless have a “special rather than… apparent meaning” (parag 5 [37]). Determining the content of this special meaning required a contextual construction, which in the Canonba case would have involved deciding the meaning of a “feral pig” in locations where these pigs were found. Priestly JA held that the pigs’ physical appearance was relevant, because this would assist in differentiating them from domestic pigs (parag 8 [37]). However, because the word “feral” had not acquired a technical meaning, a pig who lacked physical characteristics associated with feral pigs, could still be classified as feral where it was wild or untamed, or was otherwise a domestic pig who had escaped (parag 8 [37]).

By analogy, the definition of pest accepted by the Court *Brighton v Will* [1] was sufficiently wide to apply to a companion animal, which the dog could have been, because he had a collar and was microchipped. In this situation, it is arguable that the word pest had acquired a special meaning, which excluded domestic animals. Therefore, the case warranted additional evidence to verify how the word pest should be interpreted in a situation where a domestic dog was behaving badly. This was a missed opportunity for the Court to admit evidence on behavioural responses of dogs, especially when they perceive a threat and whether in these situations, normal dog behaviour should lead to the dog being categorized as a pest. This was also important given that the Supreme Court accepted the dog was docile when he was killed and his classification as a pest depended on his behaviour.

Dog aggression frequently stems from fear, including fear of other animals (page 53 [38,39]). Consequently, aggressive conduct can be a means of communicating distance-increasing signals to another animal, either in the hope that the animal submits, or the threat goes away (page 443 [40]). If neither submission nor retreat occurs, interactions that follow might be wrongly diagnosed by human observers [39]. In such a case, a domestic dog does not cease to be a domestic dog. At worst, the behaviour can lead to consequences in accordance with the Companion Animals Act [15], that have already been discussed. Corresponding arguments apply with respect to the word “exterminate”.

### 3.3. Context: To Exterminate

As with “pest”, the Court in *Brighton v Will* [1] determined that it was appropriate to apply the dictionary meaning of “exterminate” (*Brighton v Will*, parag 72 [1]). The fact that the dictionary meaning only referred to the act of destruction and excluded consideration of the means of destruction was not seen as relevant. In the Court’s view, the decision merely hinged on whether the appellant had utterly destroyed or eliminated the dog (*Brighton v Will*, parag 72 [1]). As noted, this approach removed the context of the killing and also removed the context of wild dog management. This is a significant omission, because although pest animal management is premised on the need to kill, which in any event has been criticized [41] and the regime has increasingly conceded that ethical considerations are an important regulatory dimension.

With regard to the focus on killing scientists and veterinarians agree that as a preliminary matter, regulators, and those who deal with species’ management, should determine whether controlling unwanted species is necessary and if so, whether that equates with killing (page 6 [42]). If killing is inevitable, then stakeholders need to ensure that animal suffering is minimized (page 65 [42]).

This approach has been elaborated in a policy document, *A Model for Assessing the Relative Humaneness of Pest Animal Control Methods*”, which pinpoints that humaneness should reduce animal suffering, including the method of killing, as well as how the animal is treated prior its death [43]. In other words, humanness relates to the welfare of the animal.

These approaches have already been incorporated into a number of policy documents, including the *National Wild Dog Action Plan* and the *NSW Wild Dog Management Strategy, 2017–2021* [6]. The former confirms that humaneness is an important component, noting that the outcome of the plan will guide the implementation of a nationally-agreed framework for a strategic and risk-based approach to wild dog management, emphasising humane, safe and effective management techniques and appropriate scales for mitigating the impacts of wild dogs (page 7 [7]).

Appendix D of the plan adopts a humaneness framework based on the *Model Code of Practice for the Humane Control of Wild Dogs,* identifying acceptable and unacceptable methods of killing or control (pages 6–11 [5]). Accordingly, the framework proscribes strychnine, but not sodium fluoroacetate poison, commonly known as 1080, and approves the use of exclusion fencing, but not steel jaw traps (pages 6–11 [5]).

In a similar way, the *NSW Wild Dog Management Strategy, 2017–2021*, recognises that strategies should emphasise reducing impacts of wild dogs, rather than focusing on killing (page 8 [6]). The strategy further concedes that “[t] he general public has a legitimate interest in the humaneness, target specificity and safety of pest animal management.” (page 13 [6]).The trend towards humaneness was also evident in the earlier, Wild Dog Pest Control Order 2015, which specified that wild dogs could be eradicated “by any lawful method”, a term delineated by restrictions set out in the *Pesticides Act 1999* (NSW) and the Agvet Code (clauses 7(1) (a) and 8 [14,44]). Elsewhere, the *New South Wales Invasive Species Plan 2018–2021* integrates animal welfare and humaneness into decision making by the adoption of best practice approaches ([8]). This is understood to comprise “[m]ethods or techniques that integrate all available knowledge and research that is proven to deliver the most effective, cost-efficient and humane invasive species control…” (Appendix 2 Glossary of Terms [8]).

Two further instruments that promote humaneness are the *Standard Operating Procedure GEN001: Methods of Euthanasia* [45] and the *Game and Feral Animal Control Act 2002* (NSW) [9]. The former focusses on giving animals a quick death, specifying that euthanasia should produce “rapid loss of consciousness and death” which in small animals or animals with a soft skull may be achieved by a “sharp blow to the central skull bones” (pages 4–5 [45]). For dogs, the recommended method is a shot to the brain (page 11 [45]). The *Game and Feral Animal Control Act 2002* (NSW) authorises hunting to control a range of species, including wild dogs [schedule 3 [9] and all hunting licences are issued subject to the holder’s complying with a code of practice (*Hunters’ Code of Practice*) (sections 15, 16, 19, 24 [9]; schedule 2, clauses 5 and 7 [10]).Amongst other things, the Code stipulates that to avoid animal suffering, hunters must kill animals humanely, including taking reasonable steps to locate and kill wounded animals (schedule 2, clauses 5 and 7 [10]).

These instruments provide context for a regime which rests on control and eradication, but which also aims to bridge the gap between the reality of animal killing and the need to ameliorate animal suffering. As such, the instruments raise important questions about how animal welfare interacts with rules of statutory interpretation and, in particular, how the use of non-legislative instruments assists in identifying the intention of parliament. Rothman J indirectly touched on the importance of context when he noted that.

There can be no doubt that the intention of the legislature is to prevent unjustified animal cruelty and it has defined “unjustified” by the terms of s 530(2) of the Act. However, the implementation of that purpose is not easy to reconcile with the words that are used in the section (*Brighton v Will*, parag 117 [10]).

His Honour had trouble reconciling the intention of parliament with the text of Section 530 because the section creates an offence for serious cruelty, yet it also facilitates justification of that cruelty by the defences set out in Section 530(2). As discussed, the challenges his Honour faced partly stem from a textual interpretation of the legislation, but they also derive from the welfare underpinnings of animal law, which balance animal suffering against human uses for animals. The next part of this article evaluates the propensity of legal regimes to give preference to human interests, arguing that this magnifies the chasm between animal welfare and animal suffering, which is not alleviated by textual interpretations of legislation.

## 4. The Legal Chasm

There is no doubt that the dog in *Brighton v Will* [1] suffered. The veterinarian who performed the necropsy not only concluded that the dog would have been in considerable pain for a prolonged time, but also that the method of killing was inhumane (*Brighton v Will*, parag 19 [1]). Had the arguments of the prosecutor, that the dog was a companion animal, succeeded, then presumably the Supreme Court would not have classified the dog as a pest (*Brighton v Will*, parags 52, 56 [1]). Instead, Section 22 of the Companion Animals Act would have triggered inquiry into the method of killing [15]. That section allows a person to injure or destroy a dog when protecting property (*as Alice is in law (Per Elder Smith Goldsbrough Mort Ltd v McBride* (*1976*) *2 NSWLR 631* [46]); however, any actions must be “reasonable and necessary” (Section 22(2) [15]). In addition, a person must take reasonable steps to ensure that injured dogs receive treatment and if a dog must be killed, this should be carried out quickly and without unnecessary suffering (Sections 22(7)(a) and 22(10) [15]).

Yet, according to the decision in *Brighton v Will* [1], a dog who is a pest can be killed or exterminated by whatever means, as long it is not for the purpose of gratuitous cruelty. To secure a conviction the prosecution needs to demonstrate that the dog was killed with the subjective intention of inflicting pain. The prosecution in *Brighton v Will* [1] was unable to do so, because the Court accepted that the appellant killed the dog to protect Alice, and not for “some perverse… desire to witness the [dog’s] suffering” (*Brighton v Will*, parags 80 and 100 [1]). Such an approach, however, ignores the fact that a dog classified as a pest has the same level of sentience and capacity for suffering as a companion dog and that much animal suffering occurs without the intention of inflicting gratuitous cruelty (page 163 [47]).

This is apparent in both POCTA and Section 530(1) of the Crimes Act, which operate under a similar premise, establishing offences for cruelty, but then providing a range of exemptions that vitiate those offences (Sections 13, 15, 16 and 24 [3]; Sections 530(1) and s 530(2)(b) [2]). As such, anti-cruelty regulation is “heavily qualified”, equipping violators with the prospect of rationalising cruelty in a wide variety of circumstances (page 357 [48]). This approach is not unique to NSW, with other jurisdictions, including those in the United States of America (5Miss. Code Chapter 77 s 2918 (Rev. ed. 1880), having at various times used terms that allow cruelty to be justified, “unlawfully and maliciously”, “cruelly beat, abuse, starve, torture or purposely injure”; and in South Australia, “unlawfully and maliciously” [49].

Courts have interpreted these words as proffering leeway to determine the subjective purpose for the act or omission amounting to cruelty. In *Stephens v State (1888) 65 Miss 329*, a case from the United States of America, the Court found Stephens not guilty of animal cruelty for shooting four pigs who trespassed onto his land, because he was protecting his crops. By way of contrast, *in R v Menard* (1979) 43 CCC(2d) 458 (Que CA), the Court held that the use of car exhaust to generate carbon monoxide gas to euthanize animals amounted to cruelty, because during the two minutes it took for the animals to die, they endured burns and other pain and suffering. The Court found that the animals’ suffering could have been eliminated by a relatively straightforward and inexpensive modification to the procedure. In an analogous way, the Supreme Court of South Australia Court in *Charlton v Crafter* (1943) SASR 158 [34] held that the use of strychnine to kill stray cats amounted to animal cruelty. In that case, Morgan, who owned a catering business, fed the cats, encouraging them to remain in the area because they caught rodents. Charlton, however, regarded the cats as a nuisance and poisoned them. Although the cats were stray, and as such “less domesticated” than other cats, the court nevertheless found that they were kept for a domestic purpose and were thus protected by anti-cruelty legislation page 160 [34].

These cases demonstrate at least two things. First, that through the mechanism of statutory interpretation, courts can narrow or widen the scope of cruelty in a legal context. However, as already argued, in pest animal management, legislation is only Animal Control Regulation of the regime. This calls for statutory interpretation to have recourse to the widest range of materials to determine accurately parliament’s intention. Second, a related, and perhaps a more fundamental issue, stems from the basis of animal welfare, which is determined by anti-cruelty offences. In this situation, the legal chasm between animal welfare and animal suffering is magnified by Courts’ interpretations of *justifiable* cruelty. Law and policy tolerate animal cruelty where it aligns with human purposes, subsuming animal pain and suffering within the notion of how much suffering is deemed acceptable. The difficulty, however, is that a great deal of animal cruelty can be justified (page 163 [47]) thus the underlying issue hinges on the extent to which the law condones cruelty. The greater the acceptability of cruelty, the greater the law’s complicity in separating animal suffering from animal welfare. This is a perverse outcome, given that animal welfare is based on animal sentience, yet the gulf is almost inevitable, given the utilitarian underpinnings of the welfare paradigm.

The ethical principles of utilitarianism are founded on happiness, or “utility” [50]. With respect to animals, its use is attributed to Jeremy Bentham (1748–1832), although he drew on the works of earlier Enlightenment thinkers, including David Hume (1711–1776) and Francis Hutcheson (1694–1746), whose ideas in turn can be traced to Greek philosophers, such as Epicurus (341-270BC) [50]. Bentham’s treatise, *An Introduction to the Principles of Morals and Legislation*, contains a lengthy footnote, adopting utilitarian principles to condemn the law’s disregarded for animal suffering (pages 143–144 [51]). Some two hundred years later, Peter Singer took up the baton, in his seminal work *Animal Liberation*, concluding that if a being suffers there can be no moral justification for refusing to take that suffering into consideration: “No matter what the nature of the being, the principle of equality requires that its suffering be counted equally with the like suffering—insofar as rough comparisons can be made—of any other being” (page 8 [52]).

Although the phrase “no moral justification” appears to provide an overriding veto in favour of animal sentience, in practice this is not the case.

Utilitarianism is also a consequentialist ethic, so that the rightfulness or wrongfulness of an act or omission is judged by its consequences; however, these consequences are also moderated by the parties’ interests (page 15 [53]). This means that decision makers must balance animal sentience against the utility of animals to humans. Therefore, while human and non-human interests alike must be taken into account, humans may eschew animal interests allowing the probity of an act or omission to be judged according to human interests (pages 28 and 30 [54]; pages 20–21 [52]). The proviso, however, is that animal suffering needs to be justified. As discussed, this requires the suffering to be necessary, and/or reasonable and not inflicted maliciously or gratuitously (pages 8–9 [55]).

The practical difficulties of converting this ethic into a workable animal welfare regime were recognised by Rothman J in several places in *Brighton v Will* (*Brighton v Will*, parags, 60, 117 [1]). In addition to challenges identified in establishing the intention of parliament, his Honour also considered how to interpret the exceptions to animal cruelty in Section 530(2) of the Crimes Act:
If “intention of inflicting severe pain” was construed to mean with the purpose of inflicting severe pain, then a number of the exemptions from criminal responsibility contained in s 530(2)(b) of the Act would be rendered otiose (*Brighton v Will*, parag 126 [1]).

His honour reflected that legislation which simultaneously proscribes cruelty, but also permits it for pursuits ranging from “routine agricultural or animal husbandry activities, recognised religious practices or the extermination of pest animals or veterinary practice” was contradictory (*Brighton v Will*, parag 106 [1]). Although he accommodated the apparent contradiction by holding that the intention of parliament was to prohibit unjustified animal cruelty, he nevertheless reiterated that “the implementation of that purpose is not easy to reconcile with the words that are used in the section” (*Brighton v Will*, parag 117 [1]). The court is clearly grappling with the fact that cruelty varies according to the circumstances and animals’ relationship to humans (page 9 [55]). This is a fundamental challenge of the welfare paradigm, particularly in a legislative milieu, but it is also a challenge which to some extent can be mitigated by a contextual interpretation of statutory provisions.

## 5. Conclusions

In *Brighton v Will* [1], Rothman J noted that the legal question was not whether the conduct of the appellant was reprehensible, but whether it was criminal (*Brighton v Will*, parag 131 [1]). This observation encapsulates two challenges: a narrow challenge concerning how to interpret Section 530 of the Crimes Act; and, a broader challenge relevant to making sense of animal welfare within the framework of animal cruelty. These challenges stem from the use of anti-cruelty legislation as a means of converting utilitarianism into a legal concept. In balancing animal suffering against the utility of animals to humans, the law inevitably aims to reduce animal suffering to a level society can accept, not necessarily a level that is commensurate with animal sentience. Indeed, the majority of animal suffering occurs because of human needs or wants (page 163 [54]) resulting in the relationship between animal suffering and animal welfare that is awkwardly captured by anti-cruelty legislation. In these cases, animal suffering ceases to be cruel if it fulfills a human interest, becoming instead an issue of animal welfare, but one that is obscured by the inquiry into whether the cruelty is justifiable.

This chasm is not helped by textual interpretations of legislation which ignore how regimes operate in practice. In *Brighton v Will* [1], a contextual interpretation would have given credence to non-legislative instruments, resulting either in the dog not being classified as a pest, and/or the admission of material regarding humaneness in pest animal management. One or the other of these approaches could have led to a different result. The fact that this did not occur leaves the question open whether parliament could have intended that a provision designed to deal with serious animal cruelty could also excuse the use of a pitchfork and mallet to kill a dog over a prolonged length of time.

Underlying these problems are questions relevant to how legal regimes integrate law, science and ethics. In a legal context, the answers to these questions influence statutory interpretation, admissibility of evidence and standards of proof. Additional research on how, indeed whether, law and policy achieve such integration would be helpful to improving future legislation, its interpretation and implementation.

## Figures and Tables

**Table 1 animals-10-01497-t001:** Penalties for Animal Cruelty.

Legislation	Section	Penalty
POCTA	Section 5—cruelty	Maximum penalty: 250 penalty units in the case of a corporation and 50 penalty units or imprisonment for 6 months, or both, in the case of an individual.
POCTA	Section 6—aggravated cruelty	Maximum penalty: 1000 penalty units in the case of a corporation and 200 penalty units or imprisonment for 2 years, or both, in the case of an individual.
Crimes Act	Section 530(1)—intention to inflict severe pain	Maximum penalty: imprisonment for 5 years.
Crimes Act	Section 530(1A)—reckless as to infliction of severe pain	Maximum penalty: imprisonment for 3 years.

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
