# Peer review of "Brighton v Will: The Legal Chasm between Animal Welfare and Animal Suffering"

_animals, 2020, doi:10.3390/ani10091497_

Round 1

Reviewer 1 Report

This is a well written paper with clear arguments and logics stating the challenges and controversies of animal welfare regulations. There are some minor questions which should be addressed or justified before publication. These include adding some scientific evidences that support the author’s arguments and the future application of this review.

Line 96-98: ‘In Brighton v Will, the Royal Society for the Prevention of Cruelty to Animals (RSPCA) commenced action against Brighton under the Crimes Act, due to the serious nature of the acts of cruelty.’

Please provide the reference for this sentence.

Line 164-166: ‘overall intention of section 530 was to forgive methods of killing which under other conditions would breach section 530(1); hence, the method of killing was not in issue.’

Please provide more explanations of why section 530 justify that the method of killing was not an issue?

Line 330-332: ‘whether controlling unwanted species is necessary and if so, whether that equates with killing. If killing is inevitable, then stakeholders need to ensure that animal suffering is minimized.’

Is there any principle of justifying the necessity of killing the dog (pest) and explaining the (physiological or scientific) definition of ‘minimising’ the animal suffering?

Line 424: ‘one component of the regime’.

Would you please clarify which component?

Line 448-450: ‘Utilitarianism is also a consequentialist ethic, so that the rightfulness or wrongfulness of an act or omission, is judged by its consequences, but in the context of the parties’ interests.’

Not sure the meaning of this sentence. Please rephrase.

Conclusion

Would you please add a few sentences that, based on the review, what are suggestions or studies that may improve the future legislation?

References

For any legislation, please provide links for future reference. For cited papers, please provide doi. Also, please check the reference format.

Author Response

Thanks for this - please see file attached

Reviewer 2 Report

The arguments highlight how the legislation could be revised such that it does not contradict it's own pronouncements. The court case in this study failed to consider behavioral perceptions of animals, as well as providing an informed characterization or category of these dogs. The study has implications for policy formulation on animal welfare for agricultural purposes, as pets, in captivity or in the wild. It's surprising that two court sittings would arrive at a different conclusion where acts of cruelty are so clear. Please see other comments in the text.

Author Response

Hi Reviewer 2

Many thanks for your comments - which were interesting. If you are able to assist:

Can you refer me to any scientific articles regarding your comments with respect to dog behaviour, lines 128,155 and 170 of the manuscript.  I gather that what you are saying is that where dogs exhibit normal "doggy" behaviour, that should not be taken as an indication of whether the dog is wild or domestic, or even a pest. This is a good point, because these nuances cannot be captured in dictionary definitions.

More detailed response will be provided in a few days. :)

Reviewer 3 Report

A very important contribution to the discussion on animal welfare / suffering, related legislation and its application in the legal system. Not a typical review/research paper but I am not sure if there is any more appropriate category for this kind of legislative papers in the Animals journal. It might fit better in some specialized law journal but then it would be very likely lost for academics in animal sciences. Definitely worth publishing.

Author Response

Thanks for this. Law Journals are invariably slow – up to 2 years between review and publication. Also they are not normally open access and you say, not available to interested stakeholders.

Reviewer 4 Report

The paper is highlighting an interesting question focused in a country but that could be applied worldwide, how the same dog could be considered a pest that should be exterminated (not important how) or a companion animal with the right of having an humane killing method when justified. In other words, how the capacity of being a sentient being does not depend of the subject, but of the courts.

I would suggest to change the title of the paper to something more attractive for readers, such as: The case of the extermination of a dog for being considered a pest. How to be a sentient being depends on the interpretation of a court.

This is just an idea... ;-)

Two minor comments that need to be addressed:

L118. Please, indicate here in more detail what does it mean: the dog had become docile (this is described in any way during the process?)

L412. Carbon monoxid in contrast to Carbon dioxide is considered a humane killing method because the animal is not suffering during the process. So, please, provide more information of what happened in this case. I understand that monoxide was produced inadequaly with a motor engine because of the burns, but his need to be clarified ,as the gas by itself is extremely appropriate for killing purposes but not the cheapest methods to obtain the gas.

Author Response

HI reviewer 4 - thanks for the comments. Please see file attached.
